

# Air-Sea Fluxes of $CO_2$ and $CH_4$ from the Penlee Point Atmospheric Observatory on the South West Coast of the UK

M. Yang[1], T. G. Bell[1], F. E. Hopkins[1], V. Kitidis[1], P. W. Cazenave[1], P. D. Nightingale[1], M. J. Yelland[2], R. W. Pascal[2], J. Prytherch[3], I. M. Brooks[3], T. J. Smyth[1]

[1] Plymouth Marine Laboratory, Prospect Place, Plymouth, UK PL1 3DH.
[2] National Oceanography Centre, European Way, Southampton UK SO14 3ZH.
[3] Institute for Climate and Atmospheric Science, School of Earth and Environment, University of Leeds, Leeds, UK

*Correspondence to*: M. Yang (miya@pml.ac.uk)

**Abstract.** We present air-sea fluxes of carbon dioxide ($CO_2$), methane ($CH_4$), momentum, and sensible heat measured by the eddy covariance method from the recently established Penlee Point Atmospheric Observatory (PPAO) on the South West coast of the United Kingdom. Measurements from the southwest direction (background marine air) at three different sampling heights

(approximately 15, 18, 27 m above mean sea level, AMSL) in three different periods during 2014–2015 are shown. At sampling heights $\geq$ 18 m AMSL, measured fluxes of momentum and sensible heat demonstrate reasonable agreement with their expected transfer rates over the open ocean. This confirms the suitability of PPAO for air-sea exchange measurements. We observed reductions in the air-to-sea fluxes of $CO_2$ from spring to summer in both years, which coincided with the breakdown of the spring phytoplankton bloom. At all sampling heights, mean $CH_4$ fluxes were positive, suggesting marine emissions. Higher $CH_4$

fluxes were observed during rising tides ($20\pm3$; $29\pm6$; $38\pm3$ $\mu$mole m$^{-2}$ d$^{-1}$ at 15, 27, 18 m AMSL) than during falling tides ($14\pm2$; $21\pm5$; $22\pm2$ $\mu$mole m$^{-2}$ d$^{-1}$, respectively), consistent with an elevated $CH_4$ source from an estuarine outflow driven by local tidal circulation. Based on observations at PPAO, we also estimate the detection limit of the eddy covariance $CH_4$ flux measurement to be ~20 $\mu$mole m$^{-2}$ d$^{-1}$ over hourly timescales (~4 $\mu$mole m$^{-2}$ d$^{-1}$ over 24 hours).

## 1. Introduction

Carbon dioxide ($CO_2$) and methane ($CH_4$) are two of the most important greenhouse gases in the earth's atmosphere. Over the last few decades, large efforts have gone into quantifying the impact of the ocean on the global $CO_2$ and $CH_4$ budgets. Air-sea fluxes of these gases are usually estimated via a "bulk" formula, i.e. as the product of the waterside gas transfer velocity $k_W$ and the air-sea concentration difference. Globally, the open ocean takes up approximately a quarter of the anthropogenic $CO_2$

emission (Le Quéré et al. 2015). This estimate, limited in accuracy partly by uncertainties in $k_W$, is used in global models to constrain the terrestrial $CO_2$ uptake (e.g. Manning and Keeling 2006; Canadell et al. 2007).




The shelf seas make up only a small fraction of the global oceans, but support a significant portion of global primary

productivity and draw a substantial flux of atmospheric $CO_2$ into the ocean (Chen et al. 2013). Muller-Karger et al. (2005)

estimated that the shelf seas might be responsible for as much as 40% of global oceanic carbon sequestration. In particular, the

majority of the atmospheric $CO_2$ taken up by European shelf seas is subsequently exported into the Atlantic Ocean (Thomas et

al. 2004). Compared to the open ocean, the coastal zone tends to be more spatially and temporally heterogeneous, increasing the

uncertainty in carbon flux estimates. Regions influenced by riverine outflow and anthropogenic activities can be net sources or

sinks of atmospheric $CO_2$ (Chen et al. 2013). Processes such as respiration of allochthonous (terrestrial) organic carbon inputs,

benthic-pelagic coupling, variability in surfactant abundance, and near-surface stratification are likely to have greater importance

in shallow waters. Furthermore, $k_W$ derived from the open ocean may not always be applicable to shallow waters, where waves

shoal and break more frequently. In estuaries and coastal embayments, turbulence can also be affected by tidal-flow and currents

(e.g. Upstill-Goddard 2006). Monitoring of $CO_2$ fluxes in such dynamic and variable environments necessitates a continuous,

high temporal resolution methodology (Edson et al. 2008), such as the eddy covariance (EC) technique.

Based on seawater $CH_4$ concentrations and global modeling, $CH_4$ emission from the open ocean to the atmosphere has

been estimated to be 0.4–18 Tg yr$^{-1}$, an uncertain but probably small term in the global $CH_4$ budget (Bates et al. 1996; Bange et

al. 1994; Lelieveld et al. 1998). In certain regions such as the Arctic, however, ice melt can expose underlying $CH_4$-rich waters

(e.g. Shakhova et al 2010; Kitidis et al. 2010). Enhanced mixing ratios of $CH_4$ were measured on low elevation flights over

regions of fractional ice cover and open leads in the Arctic, suggesting a large surface source (Kort et al. 2012). On a per area

basis, shelf seas, rivers, and estuaries tend to have much greater $CH_4$ emissions than the open ocean due to benthic

methanogenesis (Bange et al. 2006; Upstill-Godard et al. 2000). Global $CH_4$ emissions from coastal regions are poorly

quantified and may be influenced by processes such as riverine outflow and tidal circulations. In shallow waters, ebullition

(bubbles rising from the sediment) represents an additional and a potentially significant source of $CH_4$ (Dimitrov 2002; Kitidis et

al. 2007). Some bubbles are not fully dissolved in seawater before surfacing and this transfer to the atmosphere is not accounted

for in bulk flux calculations that use aqueous $CH_4$ concentrations.

Direct air-sea flux measurements would help to constrain $CH_4$ cycling and could also improve our understanding of $k_W$,

especially bubble-mediated gas transfer. Thus far, estimates of $k_W$ from sparingly soluble gases such as $CO_2$ and $^3$He/$SF_6$ (e.g.

Sweeney et al. 2007; McGillis et al. 2001; Nightingale et al. 2010) increase more rapidly with wind speed than those derived

from the more soluble dimethyl sulfide (e.g. Huebert et al. 2004; Yang et al. 2011; Bell et al. 2013). This divergence may be due

to bubble-mediated gas exchange resulting from breaking waves (Blomquist et al. 2006). $CH_4$ is much less soluble than $CO_2$ in

seawater and should thus be transferred even more efficiently by near surface bubbles.



We measured air-sea $CO_2$, $CH_4$, momentum, and sensible heat fluxes by the eddy covariance method at the Penlee Point

Atmospheric Observatory (PPAO) during three periods at three sampling heights: May–June 2014 (~15 m above mean sea level,

AMSL), June–July 2014 (~27 m), and April–June 2015 (~18 m). To evaluate how representative our measurements are of air-

sea transfer, we examine the influence of sampling height and wind direction on the flux footprint (Sections 3.1 and 3.2).

Covariance fluxes of momentum and sensible heat are compared to open-ocean bulk formulae predictions based on mean wind

speed and air/sea temperatures (Section 3.3). We then look at the wind direction and diel dependence in atmospheric $CO_2$ and

$CH_4$ mixing ratios (Section 4.1). Marine $CH_4$ emission has not been quantified previously by the eddy covariance technique and

here we estimate the detection limit of this measurement (Section 4.2). Focusing on the open ocean wind sector, we elucidate the

drivers for the variability in $CO_2$ and $CH_4$ fluxes (Sections 4.3 and 4.4).

## 2 Experimental

### 2.1 Environmental Setting

The Penlee Point Atmospheric Observatory (50° 19.08’ N, 4° 11.35’ W; http://www.westernchannelobservatory.org.uk/penlee/)

was established in May 2014 by the Plymouth Marine Laboratory (PML) on the South West coast of the United Kingdom for

long-term observations of air-sea exchange and atmospheric chemistry. PPAO is in close proximity to two nearby long-term

marine stations that form the Western Channel Observatory (http://www.westernchannelobservatory.org.uk). Meteorological

variables (wind, temperature, humidity, pressure), sea surface temperature (SST), salinity, chlorophyll, dissolved organic matter

etc are measured continuously from buoys stationed at L4 (50° 15.0’ N, 4° 13.0’ W) and E1 (50° 02.6’ N, 4° 22.5’ W), which are

about 6 and 18 km south of PPAO. Seawater $pCO_2$ is measured on weekly cruises to the L4 station and biweekly cruises to the

E1 station (Kitidis et al. 2012).

PPAO is situated on the exposed Rame peninsula on the western edge of the Plymouth Sound, which is primarily fed by

the Tamar estuary from the northwest and is open to the Atlantic Ocean to the southwest. South/southwest of PPAO, the water

depths increase steadily to ~8, 15, 22, and 24 m (relative to mean sea level) at horizontal distances of 100, 300, 1000, and 1300

m (www.channelcoast.org). Northeasterly wind comes over the Plymouth Sound to PPAO and is limited to a fetch of about 5

km. The fetch over water is much longer when the wind direction is between about 110° and 250° (Figure 1). Air from the

southeast is affected by pollution from the European Continent as well as shipping emissions. Air from the southwest (often

with wind speeds in excess of 20 m s$^{-1}$) encounters much less anthropogenic influence and is more representative of the

background Atlantic (see Section 4.1).

The stone PPAO building (length, width, height of 3.5, 3.5, 3.0 m) is approximately 11 m above mean sea level (see

Figure 1), mains powered, vehicle-accessible, and uses line-of-sight radioethernet to communicate with PML (6 km to the



north/northeast). A small strip of land, rocky outcrops, and a narrow intertidal zone separate the building from the sea. Southwest and northeast of PPAO, the horizontal distance to the water's edge is 30–60 m, depending on the tide. Southeast of PPAO, the distance to water is greater (about 70–90 m) due to an exposed, pointy headland. The local tidal amplitudes (semi-diurnal) are ~5 m during spring tide and ~2 m during neap tide. The intertidal zone is only sparsely covered by macroalgae

(much less than 10% by area), likely due to frequent exposure to large waves.

### 2.2 Turbulent Flux Instrumentation

During May–June 2014, a sonic anemometer (Gill Windmaster Pro) and a meteorology station (Gill Metpak Pro) were mounted on a metal pole about 1.4 m above the PPAO rooftop. A telescopic mast (retracted length of 2.8 m and fully extended length of

12.3 m; Clark Masts) was installed on top of the observatory roof (Fig. 1) on 17 June 2014. The Windmaster Pro anemometer and the meteorology station were then moved to a cross bar on top of the mast. In February 2015, another sonic anemometer (Gill R3) was installed at the same height as the Windmaster Pro, about 80 cm apart in the horizontal. The sonic anemometers measure 3-dimensional wind velocities ($u$, $v$: the two horizontal components; $w$: the vertical component) at 10 Hz (Windmaster Pro) and 20 Hz (R3). Table 1 summarizes measurement periods and corresponding sensor heights.

Two reasons motivated us to deploy the Windmaster Pro and the R3 sonic anemometers side-by-side. First, signal dropouts at high frequencies were common for the Windmaster Pro during moderate-to-heavy precipitation, which tended to coincide with strong southwesterly winds. Valid flux measurements from the Windmaster Pro, limited to mostly dry periods, may thus be biased towards low-to-intermediate wind speeds. Second, initial drag coefficient measurements from the Windmaster Pro at PPAO were lower than expected compared to published results for air-sea momentum flux. On the advice of

the manufacturer Gill (R. McKay, personal communication, 2015), we applied a bias correction to the $w$ axis of the Windmaster Pro (+16.6% for positive $w$; 28.9% for negative $w$). This correction is not necessary for the higher grade R3 anemometer, which has individually calibrated $u$, $v$, and $w$ components. Simultaneous deployments of these two anemometers allow us to evaluate the effectiveness of the Windmaster Pro correction (Section 3.3).

### 2.3 $CO_2$ and $CH_4$ Measurements

Atmospheric mixing ratios of $CO_2$ and $CH_4$ were measured by a Picarro cavity-ringdown analyzer (G2311-f) at a sampling frequency of 10 Hz ("flux mode"). The inlet to this analyzer was mounted ~30 cm below the center volume of the Windmaster Pro anemometer. An external dry vacuum pump drew sample air via a ~18 m long 3/8'' Teflon perfluoroalkoxy (PFA) tubing at a flow rate of initially ~30 L min$^{-1}$. The pump performance deteriorated over time due to constant exposure to sea salt. A high

performance particulate arrestance (HEPA) filter was installed immediately upstream of the pump in late 2014. This resulted in





a ~15 L min$^{-1}$ reduction of the main flow, which was still well within the turbulent flow regime. The Picarro instrument subsampled from the main flow via a ~2 m long ¼'' Teflon PFA tubing at a rate of ~5 L min$^{-1}$.

The presence of water vapor ($H_2O$) degrades the measurements of $CO_2$ and $CH_4$ via dilution, spectral interference and line broadening (Rella, 2010). Miller et al. (2010) and Blomquist et al. (2014) found that ambient variability in $H_2O$ mixing ratio

causes significant bias to the EC measurements of air-sea $CO_2$ flux. We followed the recommendation of Blomquist et al. (2014) and dried the sampled air using a high throughput dryer (Nafion PD-200T-24M). Set up in counter-flow mode (reflux configuration), the dryer utilizes the low pressure of the Picarro exhaust air to dry the sample air. The ambient $H_2O$ mixing ratio is typically on the order of 1% at PPAO. With the dryer inline the measured $H_2O$ mixing ratio was reduced by 5 to10-fold. The Picarro instrument measures "ambient mixing ratios" of $CO_2$ and $CH_4$ based on precisely controlled cavity temperature and

pressure. An internal, point-by-point correction by the instrument for residual humidity yields the "dry mixing ratios" ($C_{CO2}$ and $C_{CH4}$), which we use for flux computations. Air density fluctuations (i.e. Webb et al. 1980) should thus not affect our measurements. Tuned by the manufacturer prior to our first use, we checked the instrument calibration with $CO_2$ and $CH_4$ gas standards (BOC) and occasionally determined the instrument backgrounds with nitrogen gas. The mean $CO_2$ and $CH_4$ mixing ratios were not significantly different during calibration in the presence/absence of the high throughput Nafion dryer.

For the computations of $CO_2$ and $CH_4$ fluxes ($\overline{w'C_{CO2}}'$, $\overline{w'C_{CH4}}'$), a lag correlation analysis is performed hourly to determine the time delay between the instantaneous vertical wind velocity and the gas mixing ratios. Here the primes indicate fluctuations from the means while the overbar denotes temporal averaging. Most of the atmospheric variability in $CO_2$ and $CH_4$ is caused by horizontal transport, rather than the air-sea flux. Detrending the gas mixing ratios to remove low frequency variability improves the accuracy of the lag time determination. Between May and July 2014, a fairly consistent delay of 1.9±0.1

s was found between $w$ (Windmaster Pro anemometer) and $C_{CO2}$. After the installation of the HEPA filter, the delay increased to 3.3±0.1 s. Lag times derived from $w$ and $C_{CH4}$ are much noisier due to the smaller magnitude of the $CH_4$ flux. We apply the lag correction determined from the $w{:}C_{CO2}$ analysis to the $CH_4$ flux calculation.

Blomquist et al. (2010) and Yang et al. (2011) estimated high frequency flux attenuations of typically less than 5% from the same type of Nafion dryer as used in this study. Flux attenuation by the tubing itself should be negligible given the relatively

high flow. Considering the other larger uncertainties in our $CO_2$ and $CH_4$ fluxes (e.g. from ambient variability), we present the measured fluxes "as is" and do not apply any attenuation correction. $CO_2$ and $CH_4$ fluxes could not be computed between August 2014 and March 2015 due to faults in the Picarro instrument.

**3 Suitability of the Site for Air-Sea Transfer Measurements**

**3.1 Theoretical Flux Footprint**



We first use a theoretical flux footprint model (Kljun et al. 2004) to evaluate the suitability of PPAO for air-sea flux measurements. Typical values for southwesterly conditions (i.e. clean marine air) are used in the flux footprint calculations: roughness length ($z_0$) = 0.0001 m, friction velocity ($u_*$) = 0.20 m s$^{-1}$, and standard deviations in $w$ ($\sigma_W$) = 0.35, 0.26, 0.18 m s$^{-1}$ (to represent unstable, neutral, stable atmospheres). At a sampling height of 27 m AMSL (fully raised mast), the predicted upwind

distance of maximum flux contribution ($X_{max}$) is 600–1000 m and the distance of 90% cumulative flux contribution ($X_{90}$) is 1500–2600 m (the greater distances correspond to increased stability). For this set up, land/foreshore southwest of the observatory contributes to only 2–3% (stable) or 3–4% (neutral/unstable) of the cumulative flux, with the greater contributions corresponding to low tide and vice versa. The majority of the flux footprint is over waters ~20 m deep. At moderate-to-high wind speeds, the marine atmosphere tends to be near neutral, and the flux footprint is further away from the coastline. Unstable

conditions are more likely to occur under low wind speeds, during which the flux footprint shortens and may be more affected by the rocky coastline and near-shore wave breaking.

At our minimum sampling height of 15 m AMSL, the predicted $X_{max}$ and $X_{90}$ are 300–500 m and 900–1500 m, depending on stability. Land/foreshore southwest of the observatory is still only predicted to account for a small percentage of the cumulative flux (3–6%, varying with tide and stability). Southeast of PPAO where the distance to the water's edge is greater,

more terrestrial influence (5–9%) is predicted. We note that the Kljun et al. flux footprint model is developed for spatially homogeneous environments. A strong point source or sink within the footprint would have a disproportionately large influence on the flux.

### 3.2 Flux Processing and Evaluation of Wind Sectors

The coastal environment near PPAO is complex and heterogeneous in both air and water phases. Shifts in airmass and wind direction result in substantial changes in air temperature, $CO_2$, and $CH_4$ mixing ratios, complicating the interpretation of flux measurements. We choose a relatively short averaging interval of 10 minutes (as used by e.g. Miller et al 2010) to more easily satisfy the homogeneity/stationarity requirements for eddy covariance (see Appendix for flux quality control). Prior to flux computation, a double rotation (Tanner and Thurtell, 1969; Hyson et al. 1977) streamline correction is applied to wind data in

10-minute blocks. Tilt angles between the horizontal and vertical planes from the second rotation for sampling heights of 15, 18, and 27 m AMSL are shown in Figure 2. For wind blowing from the sea, the mean tilt angle is positive as air is forced upwards. The tilt angle is negative for the northwest sector due to the presence of a small hill behind the observatory building. A peak in tilt angle near 120°, more apparent at low sampling heights, is likely caused by the exposed headland in that direction. The impact of this local topography is reduced with increasing sampling height.





From the friction velocity $u_* = (\overline{u'w'}^2 + \overline{v'w'}^2)^{1/4}$ and wind speed ($U_{true}$), we compute the drag coefficient $C_D = (u_* / U_{true})^2$. Bin-averaged $C_D$ at the three sampling heights as a function of wind direction is shown in Figure 3. At 15 and 18 m AMSL, measured $C_D$ from about 80 to 150° are clearly elevated compared to open ocean values (which typically range between $0.5 \times 10^{-3}$ and $2.5 \times 10^{-3}$ depending on the wind speed; Edson et al. 2013). This is likely because a part of the flux

footprint overlapped with the coastal headland in that direction, which has a greater roughness length than the surface ocean. Likewise, high $C_D$ values between 250° and 40° are caused by land. The local headland effect to the southeast is no longer obvious at a sampling height of 27 m AMSL, when the flux footprint is predicted to be further away from the observatory. For winds blowing from the northeast and southwest, measured $C_D$ is lower and much closer to values expected for the open ocean. Northeasterly winds are relatively infrequent (~8% of the time) and limited in fetch; also the airmass from that direction is

affected by terrestrial pollution and ship emissions. We thus focus on the more frequent (~20% of the time) southwest wind sector (180–240°) for most of this paper.

### 3.3 Verification of Momentum and Sensible Heat Transfer

Here we compare the 10-m neutral drag coefficient ($C_{D10N} = (u_* / U_{10N})^2$) and sensible heat fluxes to the fairly well

established open-ocean bulk formulae predictions. The 10-m neutral wind speed $U_{10N}$ is determined using Businger-Dyer relationships (Businger, 1988) from the wind speed and air temperature at PPAO, tidal-dependent sampling height, and SST from L4. EC sensible heat flux is derived from the sonic temperature and further corrected for humidity using the bulk latent heat flux. To avoid sheltering by Rame Head to the west and near-shore processes, we limit our $C_{D10N}$ observations to a narrower wind sector of 180–220°. Figure 4 shows the relationship between $C_{D10N}$ and $U_{10N}$ from the Windmaster Pro sonic anemometer.

Also shown are the predicted $C_{D10N}$ from the COARE model version 3.5 (Edson et al. 2013) and Smith (1980). When the sensors were initially placed at 15 m AMSL, measured $C_{D10N}$ values were significantly above the open-ocean parameterizations at moderate wind speeds, likely because land/foreshore was within the flux footprint. At 18 m AMSL, the mean $C_{D10N}$ at intermediate-to-high wind speeds is similar to bulk predictions. Measured $C_{D10N}$ are sometimes elevated at wind speeds less than ~5 m s$^{-1}$, possibly due to increased flow distortion or minor land influence.

At 27 m AMSL, a limited number of $C_{D10N}$ measurements from the Windmaster Pro within the wind sector of 180–220° are available (valid flux segments N=42), which appear to be lower than the open-ocean parameterizations by about $0.2 \times 10^{-3}$. These low $C_{D10N}$ values may partly be due to remaining uncertainties in the Windmaster Pro sonic anemometer even after applying the bias correction to the $w$ axis. Our coastal measurements show that at a tilt angle of about 5°, the recommended $w$ correction increases $u_*$ from the Windmaster Pro by 6% (and increases scalar fluxes by 14%). Relative to the R3 sonic





anemometer, this reduces the low bias in the Windmaster Pro $u_*$ from 9–10% to 3–4%. The remaining 3–4% bias explains an approximate $0.1 \times 10^{-3}$ underestimation of $C_{D10N}$ by the Windmaster Pro.

Figure 5 shows a comparison between the EC sensible heat flux and the bulk sensible heat flux. The latter is computed from SST from the L4 buoy, potential air temperature and $U_{10N}$ from PPAO, and the heat transfer velocity parameterization from

the COARE model (Fairall et al. 2003). Measurement and prediction are not far from the 1:1 line at a sampling height of 27 m AMSL (slope = 0.82; $r^2$ = 0.72). A perfect agreement is not expected here, as any spatial heterogeneity in SST along the 6 km between L4 and PPAO (e.g. due to the Tamar estuary outflow) or near-surface vertical gradient in seawater temperature would contribute to the discrepancy between measured and predicted sensible heat flux. At the initial sampling height of 15 m AMSL, measured sensible heat flux is often very large and shows no correlation with the bulk flux estimate, most likely due to the

terrestrial influence within the flux footprint. At 18 m AMSL, a better coherence is observed but significant scatter remains, likely in part because the largest horizontal variability in SST is close to shore (and occupies more of the footprint at 18 m than at 27 m). Overall, our comparison of measured and predicted momentum and sensible heat fluxes suggests that data collected at a sampling height ≥ 18 m during southwesterly winds are reasonably representative of air-sea transfer.

**4 Results and Discussion**

**4.1 Variability in CO₂ and CH₄ Mixing Ratios**

Mixing ratios of $CO_2$, and $CH_4$ ($C_{CO2}$ and $C_{CH4}$) varied at PPAO depending on wind direction (Figure 6). On average between May and July 2014, $C_{CO2}$ and $C_{CH4}$ were generally higher for winds blowing from land than for winds blowing from the sea, likely due to the much greater terrestrial emissions of these gases and also different atmospheric dynamics. Mean $C_{CO2}$ and $C_{CH4}$

from the southwest sector (180–240°) are similar to "well mixed" atmospheric observations from sites such as Mauna Loa and Mace Head, consistent with the long atmospheric lifetime of these gases. Mean diel cycles in $C_{CO2}$ and $C_{CH4}$ between May and July 2014 during onshore (110–240°) and offshore (300–60°) wind flows are shown in Figure 7. $C_{CO2}$ and $C_{CH4}$ for onshore winds show little diel variability, consistent with the relatively small air-sea $CO_2$ and $CH_4$ fluxes (on a per area basis). $C_{CO2}$ and $C_{CH4}$ for offshore winds increased at night and peaked in the early morning. Nighttime wind speeds tend to be low during

offshore flow, with an average of ~3 m s⁻¹ during these months. The resultant low atmospheric turbulence favors the formation of a shallow nocturnal boundary layer, which traps surface emissions. Between about 11:00 and 20:00 UTC, $C_{CO2}$ was lower for offshore winds than for onshore winds, probably due to terrestrial photosynthesis. Diel cycles in $C_{CO2}$ and $C_{CH4}$ have been observed previously at terrestrial sites (e.g. Winderlich et al. 2014). Clear day/night differences were also apparent in the mixing ratios of oxygenated volatile organic compounds measured from the rooftop of PML (Yang et al. 2013). While not the focus of





this work, it is worth noting that the elevated atmospheric $CO_2$ and $CH_4$ in the early morning for offshore winds will influence

their air-sea fluxes in coastal waters.

**4.2 Detection Limit of $CH_4$ Flux Measurement**

In this section, we examine the eddy covariance flux detection limit of $CH_4$ and its dependence on instrumental noise as well as

ambient variability. Ten-minute segments of $CO_2$ and $CH_4$ fluxes that pass the quality control criteria (see Appendix) are further

averaged to hourly intervals. The hourly averaging reduces random noise by a factor of $\sim N^{0.5}$, where N is the number of valid

flux segments. Only hours with at least three 10-minute flux intervals are considered for further analysis.

    Blomquist et al. (2014) estimated an hourly $CO_2$ flux detection limit of $\sim$1 mmole m$^{-2}$ d$^{-1}$ for a prototype version of the

Picarro analyzer (G-1301-f) with a Nafion dryer at a wind speed of 8 m s$^{-1}$ and in a neutral atmosphere. This represents an order

of magnitude improvement over previous $CO_2$ sensors (e.g. Licor) and is lower in magnitude than the typical air-sea $CO_2$ flux.

Based on terrestrial eddy covariance measurements, Peltola et al. (2014) estimated the $CH_4$ flux detection limit using the Picarro

analyzers G-1301-f and G-2311-f to be $\sim$170 μmole m$^{-2}$ d$^{-1}$ for an averaging interval ($T$) of 30 minutes ($\sim$120 μmole m$^{-2}$ d$^{-1}$ at $T$

= 60 minutes). In comparison, the expected emission of $CH_4$ ($F_{CH4}$) based on open ocean seawater $CH_4$ concentrations is

generally less than 10 μmole m$^{-2}$ d$^{-1}$ (e.g. Forster et al. 2009) but can be significantly higher in coastal waters (e.g. Kitidis et al.

2007).

    We estimate the air-sea $CH_4$ flux detection limit using an empirical and a theoretical approach. First, following Spirig

et al. (2005), we compute the variability in the $C_{CH4}$:$w$ covariance at a time lag far away from the true lag (i.e. +300 s). During

periods of consistent southwesterly winds, the 1 σ of this "null" $CH_4$ flux is approximately 15 μmole m$^{-2}$ d$^{-1}$ at $T$ = 10 minutes.

The flux detection limit (defined as 3 σ) should thus be $\sim$18 μmole m$^{-2}$ d$^{-1}$ ($= 3 \cdot 15/6^{0.5}$) for an hourly average and $\sim$4 μmole m$^{-2}$ d$^{-}$

$^{1}$ for a daily average.

    Based on theory and scalar flux observations, Blomquist et al. (2010, 2012) attributed total uncertainty in eddy

covariance flux ($\delta F_C$) to ambient variance ($\sigma_{Ca}{}^2$) and sensor noise ($\sigma_{Cn}{}^2$):

$$\delta F_C = \frac{2\sigma_W}{\sqrt{T}} \left[ \sigma_{Ca}{}^2 \tau_{WC} + \sigma_{Cn}{}^2 \tau_{Cn} \right]^{1/2} = \frac{2\sigma_W}{\sqrt{T}} \left[ \sigma_{Ca}{}^2 \tau_{WC} + \frac{\phi_{Cn}}{4} \right]^{1/2} \qquad (1)$$

Here $\tau_{WC}$ and $\tau_{Cn}$ are the integral time scales for ambient variance and noise variance. The noise term in Eq. 1 relates to $\phi_{Cn}$, the

band-limited noise. According to the manufacturer the precision of the Picarro G2311-f is ≤ 3 ppb for $CH_4$ at a sampling rate of

10 Hz. The variance spectra of $CH_4$ during two periods of southwesterly winds are shown in Fig. 8. Variance below $\sim$0.025 Hz

largely follows the expected -5/3 slope for atmospheric transport. At frequencies above $\sim$0.025 Hz, the Picarro shows a "pink"

background noise that approximately scales to a -1/5 slope. The integrated variance from 0.025 to 5 Hz is $\sim$1.1 ppb$^2$, while the



average $\phi_{Cn}$ between 1 and 5 Hz is ~0.23 ppb$^2$ Hz$^{-1}$. Considering noise alone (i.e. $\sigma_{Ca}^2 = 0$), for a neutral atmosphere at a wind

speed of 10 m s$^{-1}$ and a sampling height of 20 m AMSL, Eq. 1 predicts an uncertainty in hourly CH$_4$ flux of 11 μmole m$^{-2}$ d$^{-1}$

(Figure 9). From the expected air-sea CH$_4$ flux, using similarity theory we can estimate the variability in $C_{CH4}$ due to air-sea

exchange in a neutral atmosphere as 3| $F_{CH4}$|/$u_*$ (e.g. Fairall et al. 2000; Blomquist et al. 2010). For $F_{CH4}$ = 2–20 μmole m$^{-2}$ d$^{-1}$

and $u_*$ = 0.3 m s$^{-1}$, this corresponds to a predicted variability of 0.006–0.057 ppb. Figure 9 shows that if the ambient variability

in $C_{CH4}$ were in this range, the hourly flux uncertainty would be dominated by sensor noise.

The observed ambient variability in $C_{CH4}$ tends to more than an order of magnitude greater than is predicted from

similarity theory, which is likely related to processes other than air-sea flux (e.g. spatial heterogeneity and horizontal

atmospheric transport). We estimate $\sigma_{Ca}^2$ as the second point of the autocovariance of $C_{CH4}$ (the difference between the first and

second points of the autocovariance equates to $\sigma_{Cn}^2$ of ~1 ppb$^2$). At PPAO, the minimum CH$_4$ ambient variability is about 0.2

ppb ($\sigma_{Ca}^2 = 0.04$ ppb$^2$), which corresponds to a predicted hourly flux uncertainty of ~20 μmole m$^{-2}$ d$^{-1}$ (Figure 9). This is close to

our earlier empirical estimate of the CH$_4$ flux detection limit above. With increasing $\sigma_{Ca}$ (i.e. more variable $C_{CH4}$), the flux

uncertainty increases substantially and becomes much greater than $F_{CH4}$, while the relative importance of $\sigma_{Cn}^2$ decreases. Thus,

we expect the 10-fold greater CH$_4$ flux detection limit estimated by Peltola et al. (2014) to be due to the higher variability in $C_{CH4}$

over land than over the sea. Over the open ocean where $\sigma_{Ca}$ in CH$_4$ is likely even lower than at PPAO, the flux detection limit

for CH$_4$ should slightly decrease.

From the analysis above, it seems that an improvement in the precision of the CH$_4$ instrument will only marginally

reduce the uncertainty in CH$_4$ flux. Blomquist et al. (2010) arrived at a similar conclusion in an analysis of air-sea carbon

monoxide flux. At present, the relative CH$_4$ flux uncertainty is best minimized by measuring in regions of large flux (i.e. high

seawater supersaturation and strong winds) and minimal ambient variability. Even under such favorable conditions, the need to

average over many hours to reduce the noise in air-sea CH$_4$ fluxes seems inevitable.

### 4.3 CO$_2$ Flux

Eddy covariance CO$_2$ fluxes measured at sampling height of 27 m AMSL and from the marine sector between June and July

2014 were generally small (see Figure 10). Diurnal land-sea breezes were common and durations of onshore winds tended to be

short during this period. CO$_2$ fluxes from the southwest (negative = into the ocean) ranged between 3 and -9 mmole m$^{-2}$ d$^{-1}$

(mean of -3 mmole m$^{-2}$ d$^{-1}$) during the relatively windy periods on 27 June and 4 July. Seawater pCO$_2$ at the L4 station ranged

between 326 and 345 μatm (mean of 337 μatm) from 9 June to 7 July 2014. The atmospheric CO$_2$ mixing ratio at L4 agrees well

with Picarro measurements at PPAO during onshore flow (Figure 10). Using the air-sea difference in partial pressure of CO$_2$

($\Delta$pCO$_2$), SST and salinity at L4, as well as wind speed at PPAO, we compute the expected air-sea CO$_2$ flux = $k_W.\alpha.\Delta$pCO$_2$,





where $\alpha$ is the solubility of $CO_2$ and $k_W$ is the gas transfer velocity from Nightingale et al. (2000) adjusted for Schmidt number. The expected air-sea $CO_2$ flux of -1 to -5 mmole m$^{-2}$ d$^{-1}$ (mean of -3 mmole m$^{-2}$ d$^{-1}$) on 27 June and 4 July are of the same magnitude as our EC measurements. The mean EC $CO_2$ flux could not be distinguished from zero in the second half of July, consistent with the increase in seawater p$CO_2$ at L4. The spring algal bloom ended abruptly in early July 2014, with chlorophyll

*a* concentration dropping from ~3 to less than 1 mg m$^{-3}$ (http://www.westernchannelobservatory.org.uk/buoys.php). The rapid warming of seawater from ~13 °C in June to ~18 °C in July aided a rapid approach towards air/sea $CO_2$ equilibrium by the middle of July 2014.

Air-to-sea $CO_2$ fluxes as substantial as -90 mmole m$^{-2}$ d$^{-1}$ were observed between April and June 2015 (sampling height of 18 m AMSL, Figure 11). For the southwest sector, the mean fluxes (standard errors) computed from the Windmaster Pro and

the R3 sonic anemometers were -19.3 (±1.4) and -23.7 (±1.4) mmole m$^{-2}$ d$^{-1}$ during this period, respectively. The reduced mean flux from the Windmaster Pro was primarily caused by signal dropouts in this anemometer during moderate-to-heavy precipitation, which tended to coincide with high wind speeds (and greater air-sea transfer). When both sonic anemometers were functional, $CO_2$ fluxes computed from the Windmaster Pro and the R3 demonstrate excellent agreement (slope of 0.98, r$^2$ = 0.95). Example $CO_2$ cospectra over about half a day from 24 April (wind speed of 8 m s$^{-1}$) and 10 May 2015 (wind speed of 6 m

s$^{-1}$) are shown in Figure 12. Minimal flux loss at high frequencies is evident, as the observed cospectra are fairly well described by theoretical fits for a neutral atmosphere (Kaimal 1972).

Hourly $CO_2$ flux (reversed in sign for clarity) during this period clearly increased with wind speed (Figure 13). Unfortunately seawater p$CO_2$ was not measured during this period for comparison. For reference, p$CO_2$ measurements from L4 in May 2014 had a mean (1 $\sigma$) of 306 (26) µatm, implying a $\Delta$p$CO_2$ close to -100 µatm. We compute the predicted $CO_2$ fluxes

using $\Delta$p$CO_2$ of -50 and -100 µatm, SST of 12.5 °C (mean from the E1 station), and the Nightingale et al. (2000) $k_W$ parameterization. During most of this period, EC $CO_2$ flux is fairly close to prediction using $\Delta$p$CO_2$ = -100 µatm. Towards late May/beginning of June, the magnitude of $CO_2$ flux appeared to be smaller at high wind speeds. A reduction in $\Delta$p$CO_2$ as occurred in 2014 could explain the declining $CO_2$ fluxes in 2015. We plan to make regular measurements of seawater p$CO_2$, SST and salinity within the flux footprint in the future, which will enable us to directly estimate the $CO_2$ gas transfer velocity.

Measured $CO_2$ flux from the southwest between May and June 2014 (sampling height of 15 m AMSL) varied from a mean (± 1 standard error) of about 40 (±8) mmole m$^{-2}$ d$^{-1}$ at night to -55 (±11) mmole m$^{-2}$ d$^{-1}$ during the day (Figure 14). Mean wind speeds were fairly similar between day and night at around 5 m s$^{-1}$ during this period. The pronounced diel variability and large magnitude of the $CO_2$ flux suggest that these fluxes were likely affected by photosynthesis and respiration from land upwind of the observatory building and/or organisms living on the foreshore. As atmosphere-land exchange of $CO_2$ can be more

than an order of magnitude larger than air-sea $CO_2$ flux on a per area basis (e.g. Goulden et al. 1996), a relatively small terrestrial



contribution to the flux footprint (>5% spatially) could significantly bias the EC measurement. At sampling heights $\geq$ 18 m AMSL, $CO_2$ fluxes show much less diel variability, as would be largely expected for air-sea transfer (Figure 14). However, the possibility of minor influence from land/foreshore on measurements at 18 m AMSL cannot be ruled out. Such local effects might explain some of the scatter in $CO_2$ fluxes at wind speeds below ~5 m s$^{-1}$, i.e. when the flux footprint was probably closer to

land.

### 4.4 CH$_4$ Flux

It is not straightforward to objectively assess the validity of our $CH_4$ flux measurements without in situ measurements of seawater $CH_4$ concentration. According to the compilation by Bange et al. (2006), typical seawater saturations of $CH_4$ range

from 110–340% in the shelf waters of the North Sea, resulting in fluxes on the order of 10 $\mu$mole m$^{-2}$ d$^{-1}$. In estuaries and river plumes, $CH_4$ saturations and hence fluxes to the atmosphere can easily be an order of magnitude higher (e.g. Upstill-Goddard et al. 2000; Middelburg et al. 2002). A strong inverse relationship between $CH_4$ concentration and salinity has been demonstrated by previous investigators (e.g. Upstill-Goddard et al. 2000), with elevated $CH_4$ concentrations found in fresher waters.

Over the three measurement periods presented here, mean EC $CH_4$ fluxes ranged between 16 and 30 $\mu$mole m$^{-2}$ d$^{-1}$ for

the southwest wind sector, with peak emissions above ~50 $\mu$mole m$^{-2}$ d$^{-1}$ (Figures 10 and 11). As with $CO_2$, during April–June 2015 the smaller mean $CH_4$ flux computed from the Windmaster Pro anemometer than from the R3 is primarily due to signal dropouts in the former during rainy, windy conditions (Table 1). The cospectra of $CH_4$ are noisier than those of $CO_2$ (Figure 12) but demonstrate the expected spectral shape. The lowest mean $CH_4$ fluxes were observed at a sampling height of 15 m AMSL, when the flux footprint was the closest to the sensor. This suggests that surface waters, rather than the intertidal zone, are the

predominant source of $CH_4$ at PPAO. In other words, the EC $CH_4$ fluxes during the low mast period in May–June 2014 are likely underestimates of air-sea transfer.

$CH_4$ fluxes from the northeast wind sector (the direction of Plymouth Sound) are on average 2–3 times higher than fluxes from the southwest (Figure 15), suggesting higher $CH_4$ concentrations in the Tamar estuary outflow than in open water. $CH_4$ fluxes from the southwest show a significant but weak relationship with wind speed (r = 0.33 during June–July 2014; r=

0.25 during April–June 2015; $p < 0.05$). The weak relationship between $CH_4$ flux and wind speed could in part be due to variable seawater $CH_4$ concentrations. $CH_4$ emissions do not obviously vary with time of day but they tend to be higher during incoming (rising) tide than during outgoing (falling) tide. In Figure 16, $CH_4$ flux from April–June 2015 is plotted against hours after low water (low tide occurs at hour zero; high tide occurs near hour six). The median, 25%, and 75% percentiles within each hour bin are also shown. The largest average $CH_4$ emissions are observed in the first ~4 hours after low tide, while $CH_4$ fluxes

during the falling tide are lower and less variable. Mean $CH_4$ fluxes were also ~50% higher during spring tide (here limited to



daily tidal amplitude > 4 m) than during neap tide (daily tidal amplitude < 3 m). These patterns are consistent with an incoming

tidal current pushing the $CH_4$-rich surface outflow from the Tamar estuary around the Rame peninsula (Uncles et al. 2015).

To further examine the influence of the Tamar estuarine plume, a 3-dimensional hydrodynamic Finite Volume

Community Ocean Model (FVCOM, Chen et al. 2003) was run for April–June 2011 with tidal forcing at the boundaries (TPXO,

Egbert et al. 2010), surface wind (Met Office Unified Model, Davies et al., 2005), surface heating (NCEP Reanalysis-2,

Kanamitsu et al. 2002), and river input (E-HYPE, Donnelly et al. 2012) at variable resolution (15 km at the open boundaries near

the shelf edge and 150 m near the Plymouth Sound). The model predicts that within 1 km south/southwest of Penlee, surface

layer (~0.2 m thick) salinity tends to be lower during rising tide (about 33.4–33.7) than during falling tide (about 33.9–

34.1). This suggests a larger freshwater outflow from the Tamar at the surface during rising tide, consistent with observations of

greater $CH_4$ emissions during these times. Natural processes other than direct air-sea gas transfer (e.g. ebullition) could also

contribute to the weak correlation between $CH_4$ flux and wind speed. Quantifications of the temporal/spatial seawater $CH_4$

distribution within the PPAO flux footprint and measurements of the pelagic/benthic cycling of $CH_4$ is essential to address this

uncertainty.

$CH_4$ emissions of a few tens of $\mu$mole $m^{-2}$ $d^{-1}$ at PPAO are generally greater than estimates for the open ocean (e.g.

Forster et al. 2009), but are lower than previous measurements over other aquatic systems. Kitidis et al. (2007) measured a $CH_4$

emission of 63 $\mu$mole $m^{-2}$ $d^{-1}$ using a floating chamber in the Ria de Vigo (a large coastal embayment), consistent with wind-

driven turbulent diffusivity models for the conditions at the time of the chamber deployment. These authors also estimated

fluxes up to 170 $\mu$mole $m^{-2}$ $d^{-1}$ during periods when the chamber was not deployed. With an open path sensor Podgrajsek et al.

(2014) recently measured $CH_4$ emission from a Swedish lake using the EC technique. Lake $CH_4$ emissions range from near zero

during the day to over 20 mmole $m^{-2}$ $d^{-1}$ at night (three orders of magnitude higher than observations at PPAO). Aircraft mixing

ratio measurements suggest that $CH_4$ emission from the partially ice-covered Arctic is 4–5 times larger than mean fluxes at

PPAO (Kort et al (2012). Our observations and estimates of the $CH_4$ flux uncertainty suggest that an EC system such as the one

employed here should be able to quantify emissions from those $CH_4$ "hot spots."

**5 Conclusions**

Air-sea fluxes of $CO_2$, $CH_4$, momentum, and sensible heat were measured by the eddy covariance technique in 2014 and 2015

from the Penlee Point Atmospheric Observatory (PPAO) on the southwest coast of the UK. Observed momentum and sensible

heat transfer from the southwest wind sector are in reasonable agreement with bulk transfer estimates at a sampling height of $\geq$

18 m AMSL. These results are consistent with theoretical calculations of the flux footprint extent. We infer that PPAO is

30    suitable for long-term, high temporal resolution measurements of air-sea exchange.





$CO_2$ fluxes demonstrate a positive dependence on wind speed and a decline in magnitude from late spring to early summer in both 2014 and 2015, coinciding with reduced air-sea $\Delta pCO_2$ driven by the demise of the spring algal bloom and a warming sea. We report the first successful EC flux measurements of $CH_4$ from the marine environment. The $CH_4$ flux detection limit is estimated to be ~20 µmole m$^{-2}$ d$^{-1}$ for an hourly average (~4 µmole m$^{-2}$ d$^{-1}$ for a daily average), which is

valuable information for planning future open ocean applications of this technique. Uncertainty in $CH_4$ fluxes is due to both instrumental noise and ambient variability in atmospheric $CH_4$ mixing ratio. Observed $CH_4$ emissions are on the order of tens of µmole m$^{-2}$ d$^{-1}$, a reasonable magnitude for an estuarine influenced coastal region. $CH_4$ fluxes are generally greater when the wind is from the Plymouth Sound than when the wind is from the southwest, suggesting an elevated source of $CH_4$ in the Tamar estuary. Mean $CH_4$ fluxes from the southwest are higher during rising tide than during falling tide. This pattern suggests an

enhanced source of $CH_4$ emission from the estuarine outflow that is affected by the local tidal circulation.

**Appendix: Quality Control on Eddy Covariance Fluxes**

Conservative quality control criteria computed from 10-minute flux averaging intervals are used to remove flux measurements during unfavorable conditions (Table A1). Periods of highly variable wind direction ($\sigma > 10°$) and positive

momentum flux are discarded on the basis of nonstationarity, which tends to occur during calm conditions or the passage of a weather front. We also reject fluxes that do not pass the statistical quality control tests for skewness and kurtosis of $w$ and integral turbulence characteristics of $\overline{u'w'}$ (Foken and Wichura, 1996; Vickers and Mahrt, 1997). Averaged valid momentum cospectra and normalized Ogives (Oncley, 1989) on 3, 5, and 10 May 2015 (R3 sonic anemometer) are shown in Figure A1. Mean wind speeds were 12, 17, and 6 m s$^{-1}$ on these three days, respectively. The Ogives approached zero at 0.0017 Hz and

approached one at 5 Hz, indicating that the 10-minute averaging interval captured the majority of the turbulent flux.

To minimize the impact of horizontal transport on $CO_2$ and $CH_4$ fluxes, we set thresholds defined by the ranges and trends in mixing ratios ($C_{CO2}$ and $C_{CH4}$) as well as the horizontal fluxes of these gases. Following Blomquist et al. (2012, 2014), we compute the horizontal fluxes as $\overline{u'C'}$ and $\overline{v'C'}$. Here $u$ and $v$ represent the along-stream and cross-stream wind velocities after double rotation. Large horizontal fluxes suggest excessive spatial heterogeneity/nonstationarity. For $CH_4$ only, we also

eliminate periods when the total variance ($= \sigma_{Cn}^2 + \sigma_{Ca}^2$) exceeds 2 ppb$^2$. Since $\sigma_{Cn}^2$ is ~1 ppb$^2$ (see Section 4.2), this equates to a $\sigma_{Ca}$ threshold of (2 ppb$^2$ − 1 ppb$^2$)$^{0.5}$ = 1 ppb and an hourly flux uncertainty of ~80 µmole m$^{-2}$ d$^{-1}$ (Figure 9). We note that this $\sigma_{Ca}$ threshold is more than an order of magnitude greater than the expected ambient variability in $C_{CH4}$ due to air-sea flux.

Both sonic anemometers show elevated noise at frequencies above 1 Hz when the relative humidity is near 100%, likely because of rain and sea spray. For computations of momentum and heat transfer, we remove moisture related artifacts by



simply discarding fluxes when the relative humidity exceeds 95%. Noise in the sonic anemometer above 1 Hz shows little correlation with $C_{CO2}$ and $C_{CH4}$, such that high humidity does not noticeably affect $CO_2$ and $CH_4$ fluxes.

### Acknowledgment

5 Trinity House (http://www.trinityhouse.co.uk/) owns the Penlee site and has kindly agreed to rent the building to PML so that instrumentation can be protected from the elements. We are able to access the site thanks to the cooperation of Mount Edgcumbe Estate (http://www.mountedgcumbe.gov.uk/). R. McKay (Gill) advised on the operation and calibration of the Windmaster Pro sonic anemometer. Thanks also to J. Stephens, J. Jury, K. Perrett, A. Staff, and B. Carlton at the Plymouth Marine Laboratory (PML) for their efforts in setting up the site and establishing data communication. R. Torres (PML) helped

10 with the FVCOM model and interpretation of the tidal data. B. Blomquist (NOAA) offered valuable advice on the setup of Picarro instrument.

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




Table 1. Summary of sampling periods, mast height above observatory rooftop and above mean sea level (AMSL), and hourly eddy covariance $CH_4$ fluxes (µmole m$^{-2}$ d$^{-1}$) for the southwest wind sector (180–240°). $CH_4$ fluxes when the sampling height was 15 m AMSL are likely underestimates of air-sea transfer because a significant portion of the flux footprint was over land (Section 3). For the last period (2015), fluxes are computed from both the Windmaster Pro and R3 sonic anemometer (shown in that order). SE indicates standard error.

|  | Sensor Height (m) |  | EC Flux | Falling Tide | Rising Tide |
|---|---|---|---|---|---|
| Time | Over roof | AMSL | Mean (SE) | Mean (SE) | Mean (SE) |
| 14 May–17 June 2014 | 1.4 | ~15 | 16 (2) | 14 (2) | 20 (3) |
| 17 June–21 July 2014 | 13.3 | ~27 | 24 (4) | 21 (5) | 29 (6) |
| 21 April–3 June, 2015 | 3.6 | ~18 | 25 (2), 30 (2) | 19 (2), 22 (2) | 33 (3), 38 (3) |

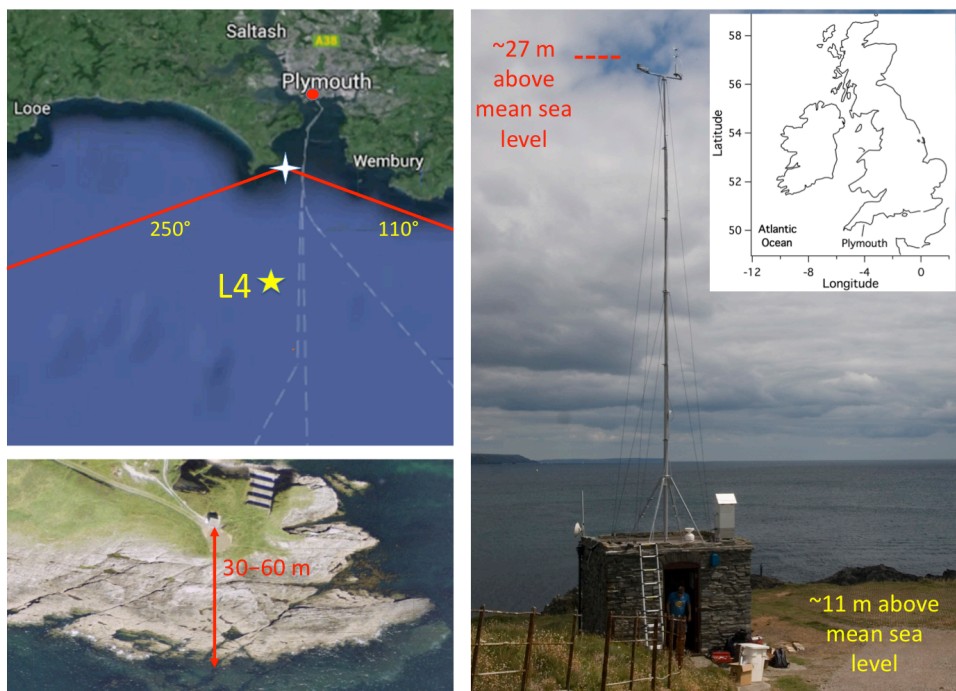

Figure 1. Location of the Penlee Point Atmospheric Observatory (white cross). The observatory is ~6 km south/southwest of the Plymouth Marine Laboratory (red dot), ~6 km north of the L4 station (yellow star), and ~18 km north of the E1 station (beyond the southerly extent of the map). White dash lines indicate commercial ferry routes. PPAO with the telescopic mast fully raised is shown on the right.




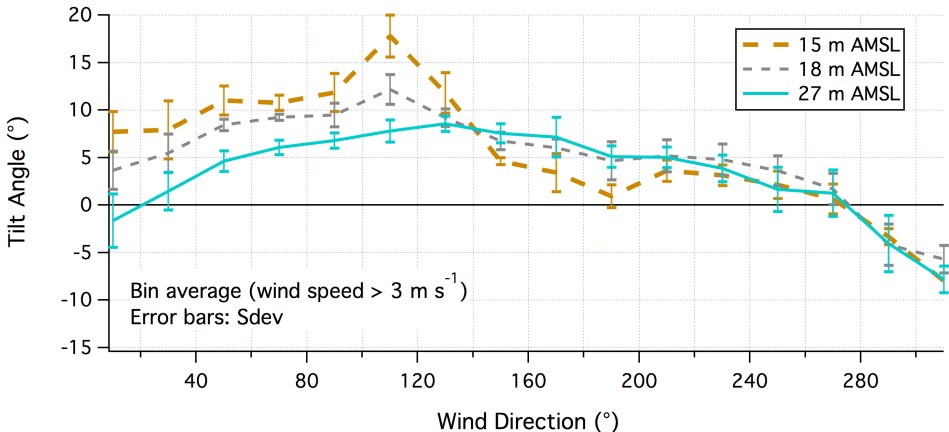

Figure 2. Tilt angle vs. true wind direction at three sampling heights. Lines represent averages (wind speed > 3 m s$^{-1}$ only) and the error bars indicate standard deviations within each wind direction bin. Wind data from the Windmaster Pro sonic anemometer.

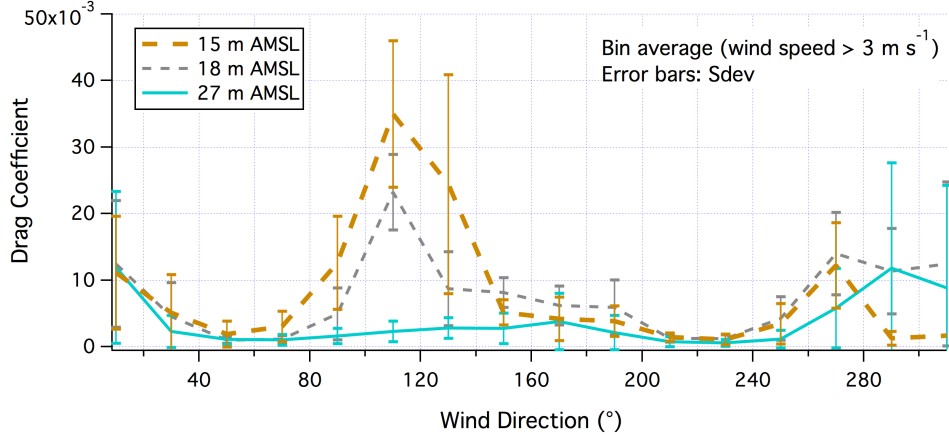

Figure 3. Drag coefficient vs. true wind direction at three sampling heights. Lines represent averages (wind speed > 3 m s$^{-1}$ only) and the error bars indicate standard deviations within each wind direction bin. Wind data from the Windmaster Pro sonic anemometer.



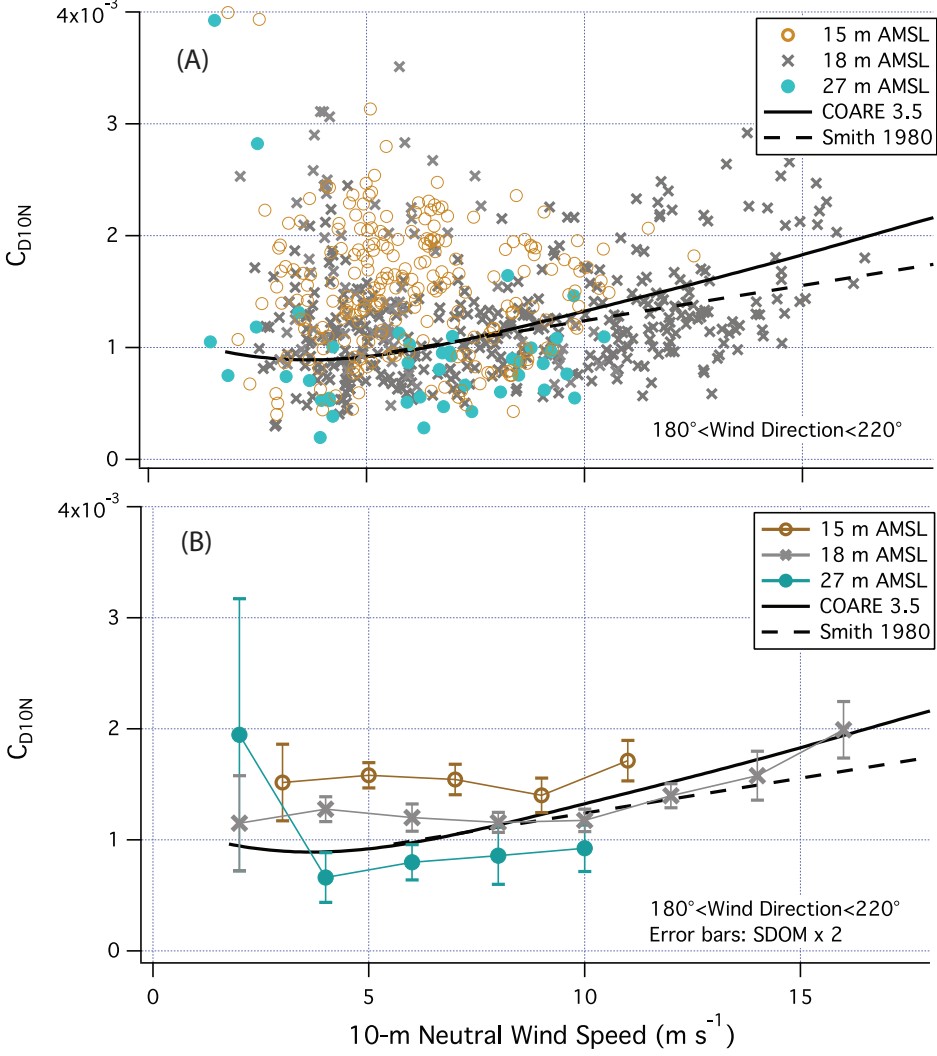

Figure 4. 10-m neutral drag coefficient vs. 10-m neutral wind speed at sampling heights of 15, 18, and 27 m AMSL. A) 10-minute EC measurements, and B) bin averages, with error bars indicating two standard errors within each wind speed bin. Wind data from the Windmaster Pro sonic anemometer. Also shown are $C_{D10N}$ parameterized from the COARE model version 3.5 (Edson et al. 2013) and Smith (1980).





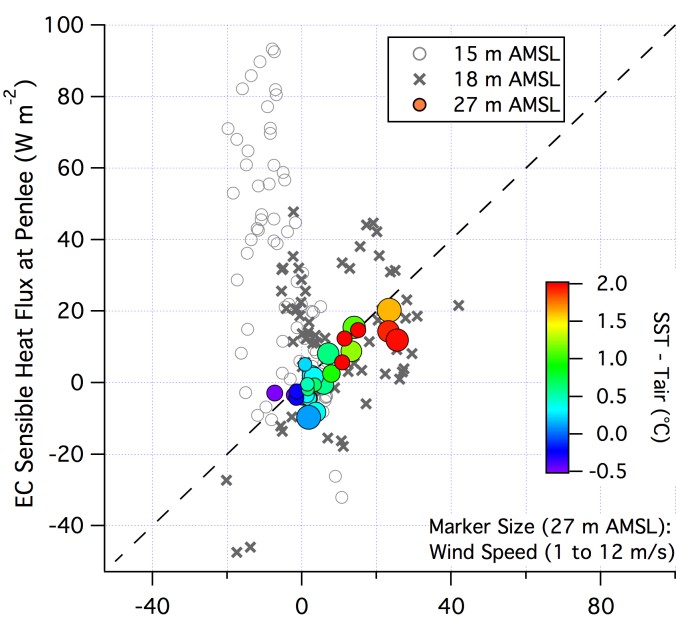

Figure 5. EC sensible heat flux vs. bulk sensible heat flux computed using SST from the L4 station. For June–July 2014 (27 m AMSL), the color-coding indicates the sea-air temperature difference, while the marker size corresponds to wind speed (1–12 m s$^{-1}$).

Figure 6. Atmospheric mixing ratios of $CO_2$ and $CH_4$ as a function of wind direction. Error bars indicate two standard errors within each wind direction bin. $CO_2$ and $CH_4$ mixing ratios were generally lower for southwesterly winds (180–240°) than for northerly wind sectors.



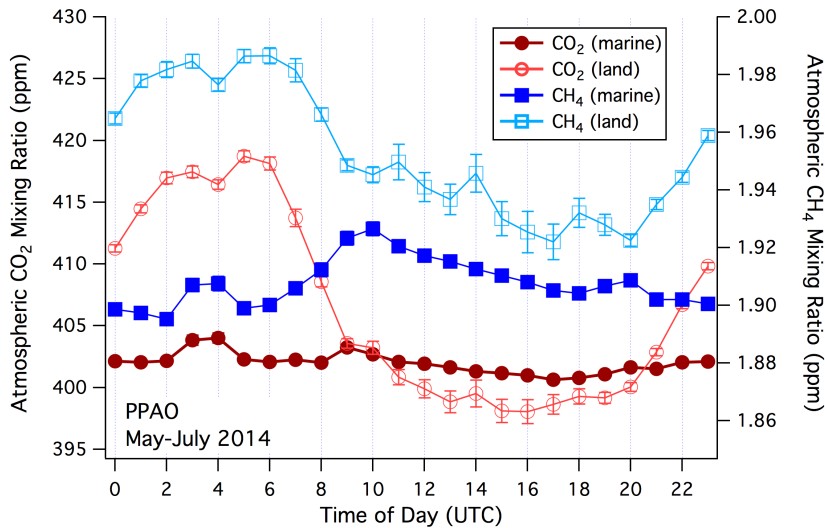

Figure 7. Mean diel cycles in the mixing ratios of $CO_2$ and $CH_4$. Error bars indicate two standard errors within each hour bin. Diel variability for both gases is small during onshore flow (marine winds, 110–240°). Mixing ratios of $CO_2$ and $CH_4$ during offshore flow (wind from land, 300–60°) increase at night and peak in the early morning.

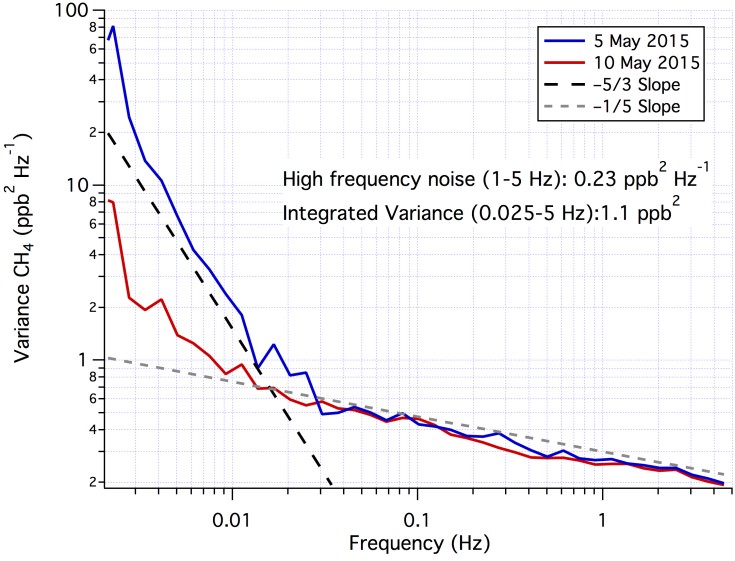

Figure 8. Variance spectra of $CH_4$ on two days of southwesterly winds. Variance at frequencies above ~0.025 Hz is dominated by noise, while ambient variability accounts for most of the low frequency variance.



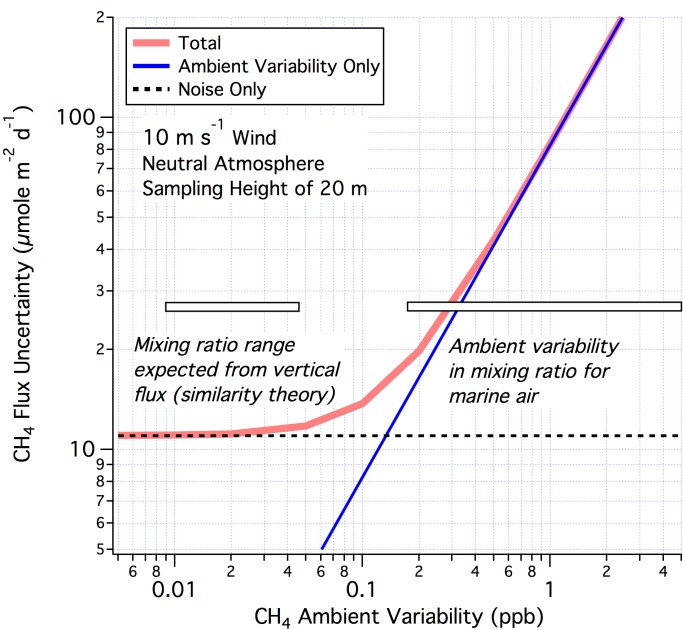

Figure 9. Estimated uncertainty in hourly averaged EC flux of $CH_4$. Typical observed and predicted (based on similarity theory
for the open ocean) values of the ambient variability in $CH_4$ mixing ratio are shown by the horizontal bars.





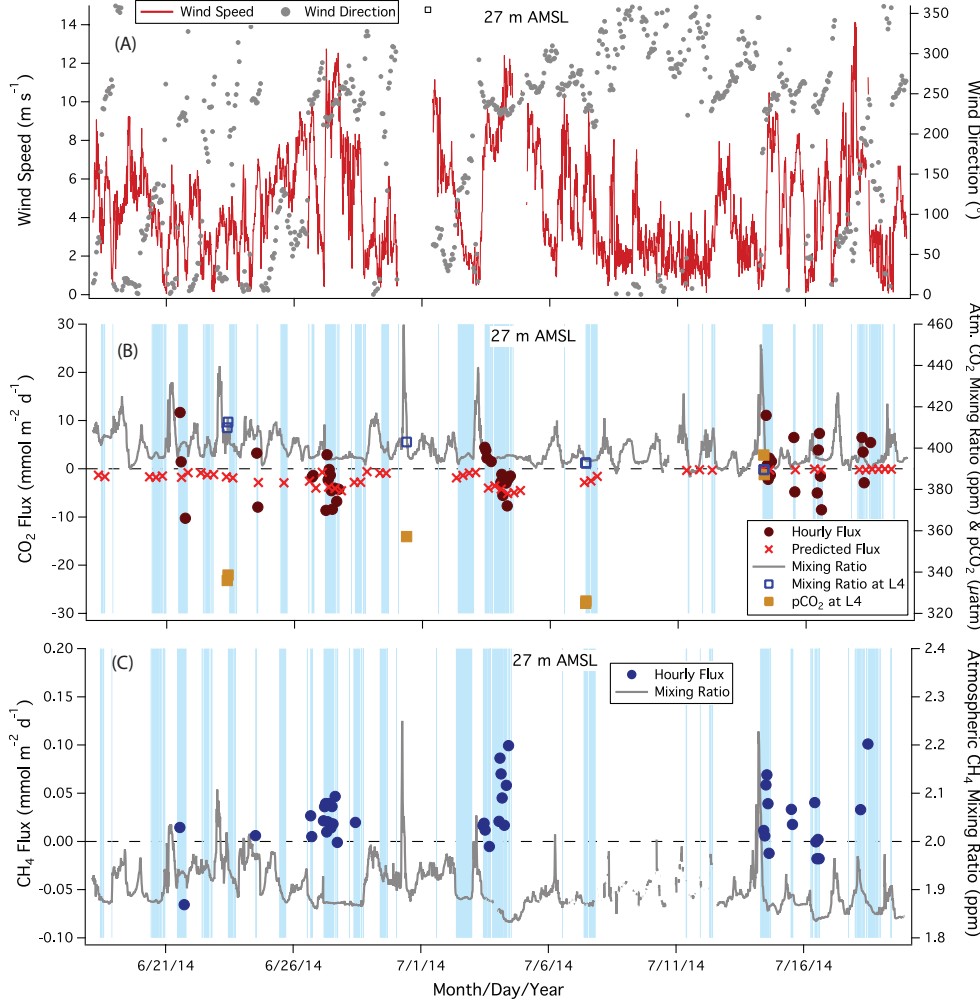

Figure 10. Time series of A) wind speed and direction, B) $CO_2$ flux and mixing ratio, and C) $CH_4$ flux and mixing ratio during June–July 2014 (sampling height of 27 m AMSL). Cyan shading indicates onshore winds. Fluxes are limited to the southwest wind sector only. Also shown are $pCO_2$ and atmospheric $CO_2$ mixing ratio from the L4 station. Negative $CO_2$ fluxes on the order of a few mmole m$^{-2}$ d$^{-1}$ were observed during the windy periods on 27 June and 4 July. By late July, observed $CO_2$ fluxes were indistinguishable from zero, consistent with near saturation of seawater $pCO_2$ at the L4 station. $CH_4$ flux has a positive mean, suggesting sea-to-air emission.





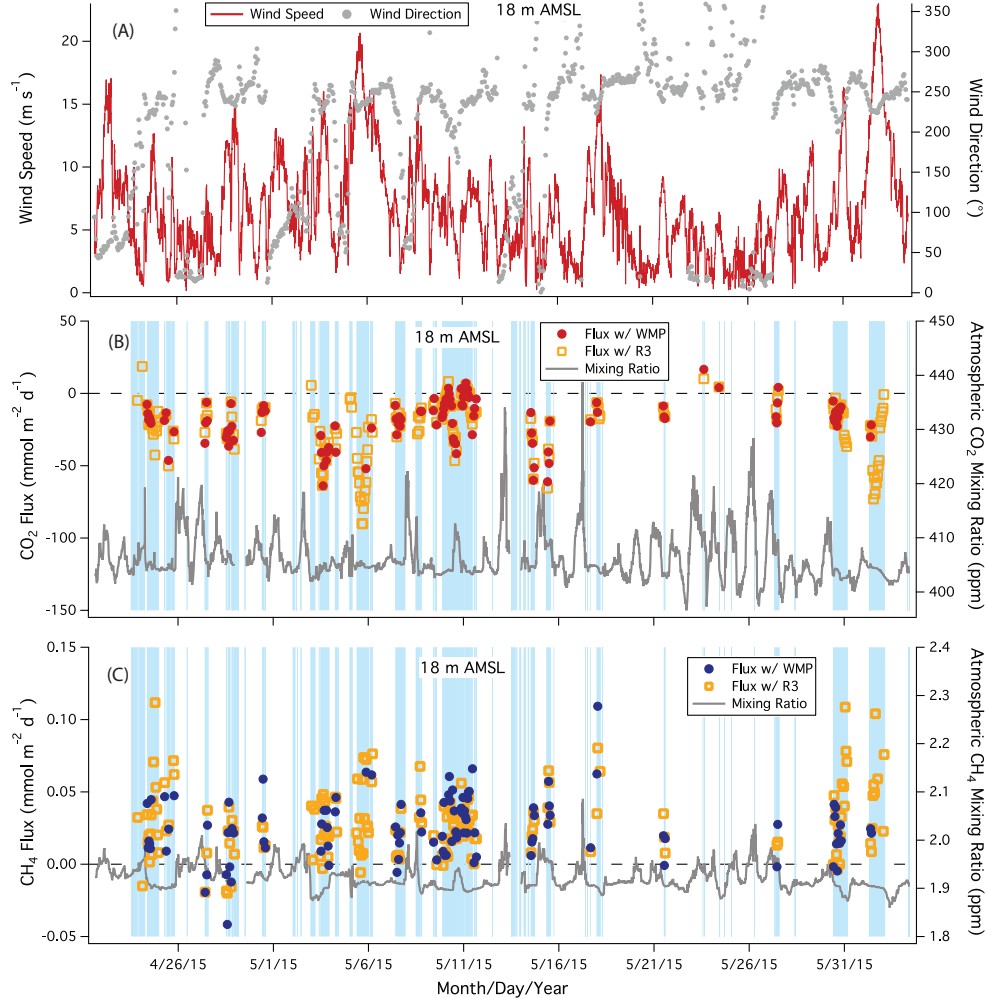

Figure 11. As Fig. 10, but during April–June 2015 (sampling height of 18 m AMSL). Fluxes were computed from both the Windmaster Pro and the R3 sonic anemometers. Large air-to-sea flux of $CO_2$ is observed during high wind speed events, while $CH_4$ flux is almost always positive.



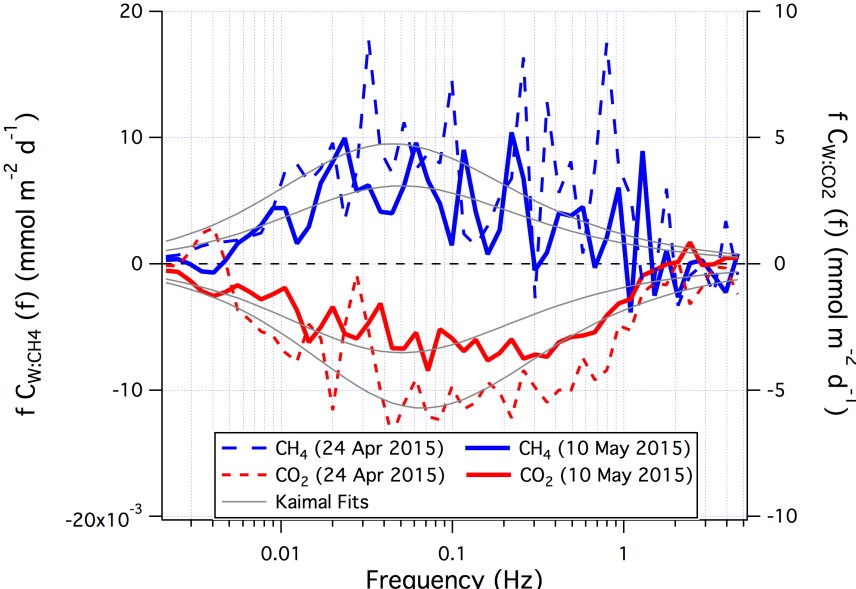

Figure 12. Mean $CO_2$ and $CH_4$ cospectra over about half a day from 24 April (wind speed of 8 m s$^{-1}$) and 10 May 2015 (wind speed of 6 m s$^{-1}$). Measurements were made at 18 m AMSL and from the southwest direction. Theoretical spectral fits (Kaimal) are also shown.

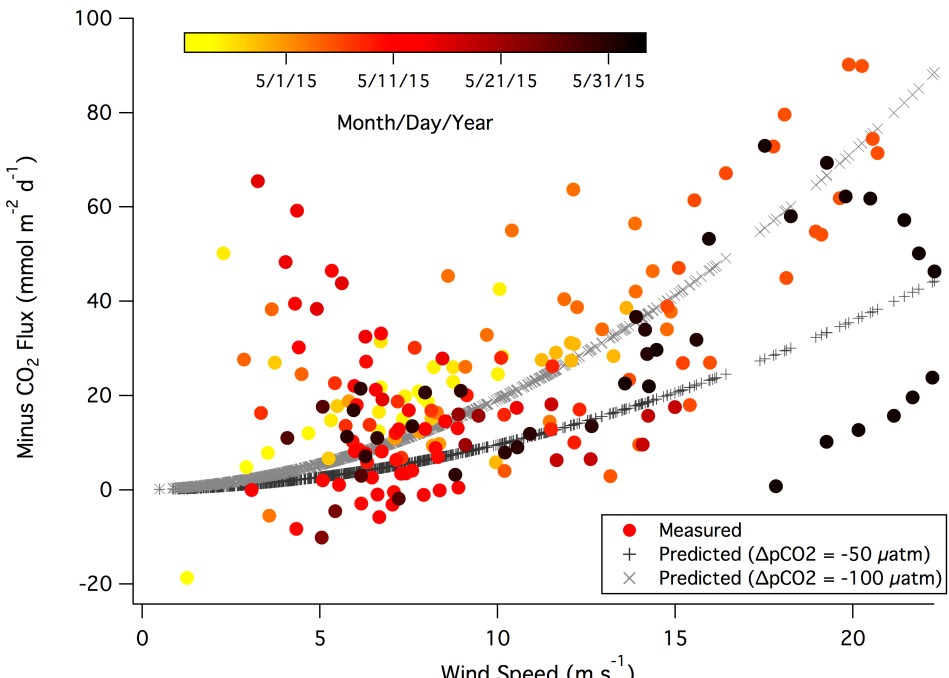

Figure 13 Relationship between $CO_2$ flux (R3 sonic anemometer; reversed in sign) and wind speed during April–June 2015 (sampling height of 18 m AMSL). Predicted $CO_2$ fluxes assuming $\Delta pCO_2$ of -50 and -100 µatm are also shown.





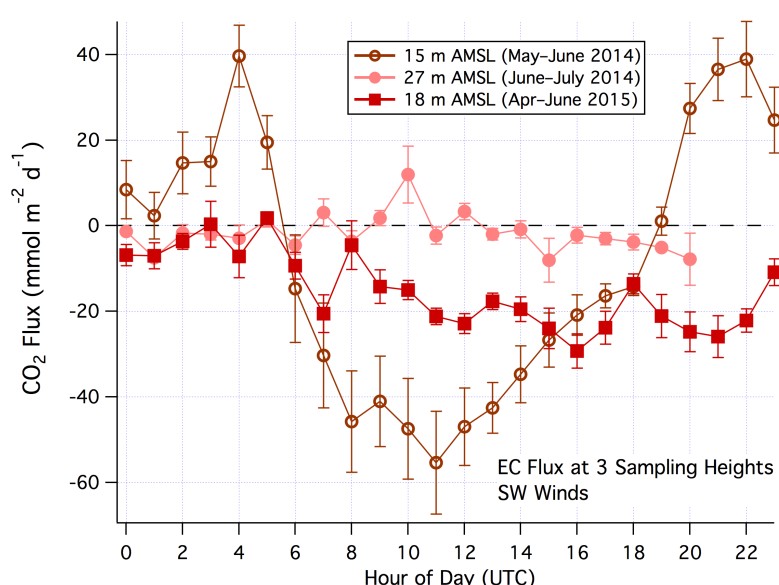

Figure 14. Diel variations in $CO_2$ fluxes at three sampling heights for southwesterly winds (180–240°). Error bars correspond to standard errors within each hourly bin. At a sampling height of 15 m AMSL, large diel variability in $CO_2$ flux was observed most likely due to a local, terrestrial influence. Fluxes measured at ≥ 18 m AMSL exhibit much less diel variability.

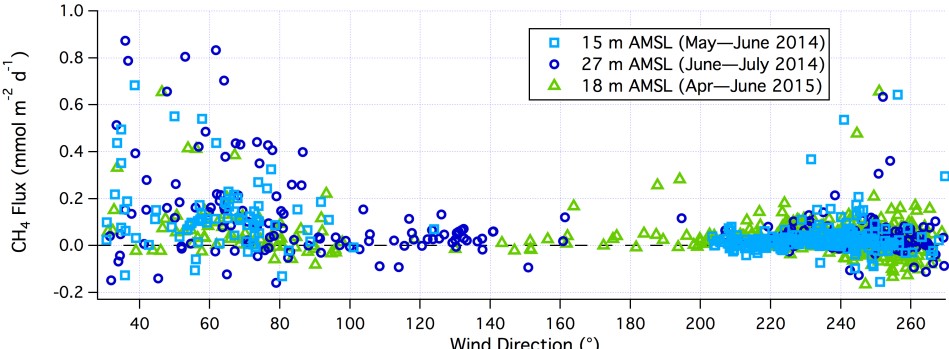

Fig. 15. Hourly $CH_4$ flux as a function of wind direction at all three sampling heights. Larger $CH_4$ emissions are generally observed when winds are from the northeast (direction of Plymouth Sound) compared to from the southwest (open water), likely due to elevated seawater $CH_4$ concentrations in the estuarine outflow.

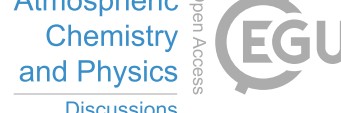



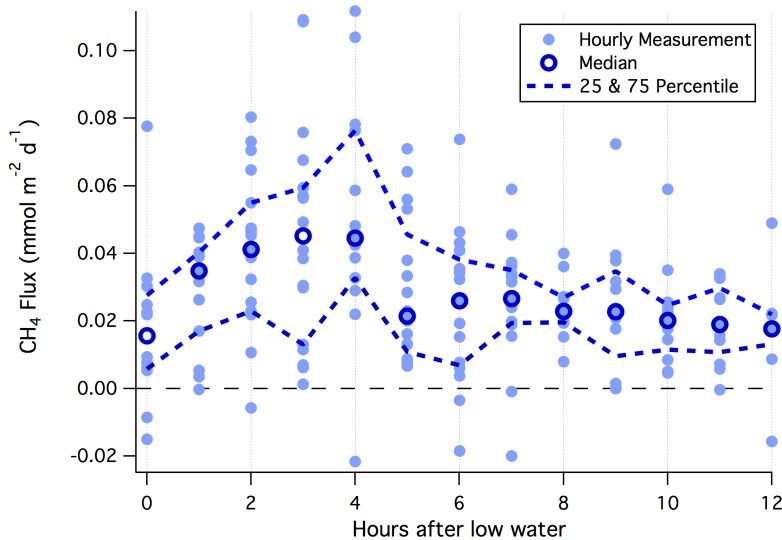

Figure 16. Hourly $CH_4$ flux from the southwest wind sector (R3 sonic anemometer) vs. hours after low water (18 m AMSL). Elevated $CH_4$ emission is observed in the first ~4 hours after low tide, consistent with an enhanced source of $CH_4$ in the Tamar estuarine outflow driven by the local tidal circulation.

35

40





Table A1. Filtering criteria (within 10-minute averaging intervals) for quality control of eddy covariance fluxes. These criteria are shown for the southwest air sector only (180°<Wind direction<240°). The right column indicates the percentage of valid flux data that data that satisfy the filtering criteria by each stage of the quality control sequence.

| | Criteria | Purpose | Percentage Passed |
|---|---|---|---|
| Wind | $\sigma$ in wind direction <10° | Choose constant wind direction | 93 |
| | Negative momentum flux | Check wind profile | 92 |
| | Pass skewness, kurtosis, and integral turbulence characteristics tests | Satisfy stationarity of wind | 88 |
| $CO_2$ & $CH_4$ | No gap in Picarro data | Verify Picarro data | 92 |
| | Valid wind | Verify physical flux | 81 |
| $CO_2$ only | $C_{CO2}$ Range < 5 ppm | Satisfy stationarity of $CO_2$ | 79 |
| | $|C_{CO2}$ Trend$|$ < 10 ppm hr$^{-1}$ | Satisfy stationarity of $CO_2$ | 75 |
| | $|$Horizontal flux$|$ <500 mmole m$^{-2}$ d$^{-1}$ | Satisfy stationarity of $CO_2$ | 74 |
| $CH_4$ only | $C_{CH4}$ Range < 20 ppb | Satisfy stationarity of $CH_4$ | 80 |
| | $|C_{CH4}$ Trend$|$ < 20 ppb hr$^{-1}$ | Satisfy stationarity of $CH_4$ | 75 |
| | Total variance <2 ppb$^2$ | Reduce flux uncertainty | 74 |
| | $|$Horizontal flux$|$ <0.4 mmole m$^{-2}$ d$^{-1}$ | Satisfy stationarity of $CH_4$ | 72 |
| $C_{D10N}$ & sensible heat | 180°<Wind direction<220° | Choose least sheltered wind sector | 72 |
| | Relative humidity < 95% | Remove moisture related noise | 67 |

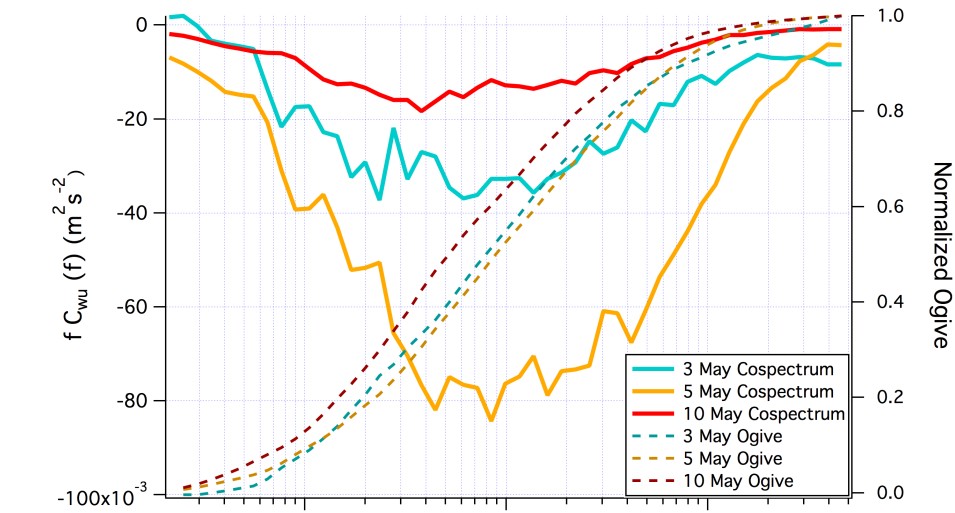

Figure A1. Mean momentum cospectra and normalized Ogives on 3, 5, and 10 May 2015 (R3 sonic anemometer). Mean wind speeds were 12, 17, and 6 m s$^{-1}$ on these three days, respectively.