# Peer review of "Air-Sea Fluxes of CO2 and CH4 from the Penlee Point Atmospheric Observatory on the South West Coast of the UK"

_Atmospheric Chemistry and Physics, 2015_

## Referee Comment (RC1) · Anonymous Referee #1 · 1 Apr 2016

The manuscript focuses on the air-sea flux of CO2 and CH4 using EC data. In particularly marine EC data of CH4 are previously mainly non-existent making the work highly interesting and worth publishing. There are, however, some major problems needed to be addressed before publication. The manuscript is very long, includes many different components and would benefit from being significantly shortened. The new and unique aspect of the manuscript is the marine CH4 fluxes and the paper would benefit from a much narrower focus. The CO2 analysis gives some numbers of the CO2 exchange, but as the water-side measurements are very limited and their representativity for the EC data highly questionable, this aspect of the paper does not bring much additional information compared to existing literature on air-sea CO2 exchange. The authors con-

clude that the site is suitable for long-term high temporal resolution measurements of air-sea exchange. For such a conclusion a much more thorough analysis is required. The site might be suitable for air-sea exchange representing coastal conditions, how representative the data are for undisturbed non-coastal air-sea exchange is not clear from the present analysis. Some more specific questions: Section 2.1: For EC measurements also the upwind topography is of importance, it is not clear from the text or figures how steep the topography is north of the site or how this might influence the measurements. Section 2.2: The reason for the bias correction of the Windmaster Pro is very unclear and need to be explained. Section 2.3: Is the signal dried in the Picarro (how does the low pressure of the Picarro exhaust give a dry signal)? Page 11: How is the wave field influenced by the coast and how would this influence the drag coefficient? Page 13: SST measured very far away and there is no information at what depth the SST is measured. SST at this distance is probably not very representative for the flux footprint and this will most likely have a large impact on the bulk calculated sensible heat flux in Figure 5. Page 14: "different atmospheric dynamics" what does this refer to? Page 17: The 10 fold greater detection limit estimated by Peltola compared to the present study is explained by the higher variability over land than over sea. To me the variability over sea (in Figures 10 and 11) also seems relatively large.

---

## Referee Comment (RC2) · Anonymous Referee #3 · 2 Apr 2016

Review of Yang et al., *Air-Sea Fluxes of CO₂ and CH₄ from the Penlee Point Atmospheric Observatory on the South West Coast of the UK*

**Summary and General Comments:** Yang et al present the first eddy covariance measurements of air-sea exchange of methane, a significant advance in the field. The manuscript outlines the detection limit for the flux measurements using a Picarro cavity ringdown instrument. In addition, Yang et al present coincident $CO_2$ flux measurements to make the argument that the Penlee Point Atmospheric Observatory (PPAO) is a uniquely situated for EC measurements of air-sea exchange in coastal regions.

In general, the paper is well written and is of appropriate length for ACP. The novel elements of this manuscript center around the $CH_4$ EC flux assessment. I recommend the paper to be published after the authors attention to the following minor comments.

**Specific Comments:**

Page 1 Line 15: Perhaps define quantitatively "reasonable agreement" as this forms the foundation for the argument of PPAO as a site for future air-sea exchange measurements.

Page 1 Line 20: Why are the fluxes listed in order of 15, 27, 18 m.

Page 1 Line 23: I encourage the authors to remove the "~" for the detection limits. This should be a calculation where the detection threshold is defined for a specific averaging time, not an approximation. If there is unconstrained uncertainty, perhaps state an upper limit?

Abstract: The abstract sells short the novel elements of the CH4 air-sea flux determinations. I encourage the authors to conclude the abstract with a more definitive statement that puts these new measurements in the content of what was known prior.

Page 3 Line 2: The abstract gives the impression that fluxes were determined at a range of altitudes at the same time, however it is clear here, that the range in altitudes also corresponds to a range in sampling periods. This should be noted in some fashion in the abstract?

Page 4 Line 20: More detail on the bias correction should be included here. Is this a general result for all windmaster pro's? Is the value set by comparison to the G3? Or was this number from Gill?

Page 5 Line 1: Is the 5 slpm, 2m subsampling line still turbulent? If not, how important is this?

Page 5 Line 9: Why is "ambient mixing ratios" in quotations? Presumably this refers to ambient absolute humidity?

Page 12 Line 8: What is the future prospective for making continuous dissolved SW measurements of $CH_4$ or other gases at this site? This seems critical for the future success of this site. Can this be done at L4?

Page 13 Line 9: Given that the primary focus of this paper is on air-sea exchange in the shelf region, I think it is most appropriate to state here that PPAO is " … high temporal resolution measurements of air-sea exchange in shelf regions."

---

## Author Comment (AC1) · 18 Apr 2016

Author Comment with regard to:

**Air-Sea Fluxes of CO2 and CH4 from the Penlee Point Atmospheric Observatory on the South West Coast of the UK**

by M. Yang et al.

18 April, 2016

Many thanks for the thoughtful *comments and suggestions from Anonymous Referee #1*. Below we present each comment (in *italic*), followed by our reply. All of our replies are incorporated into the revised manuscript where appropriate, unless indicated otherwise.

**Anonymous Referee #1**
*The manuscript focuses on the air-sea flux of CO2 and CH4 using EC data. In particularly marine EC data of CH4 are previously mainly non-existent making the work highly interesting and worth publishing.*
We are very glad to hear that the referee found our contribution valuable.

*There are, however, some major problems needed to be addressed before publication. The manuscript is very long, includes many different components and would benefit from being significantly shortened. The new and unique aspect of the manuscript is the marine CH4 fluxes and the paper would benefit from a much narrower focus. The CO2 analysis gives some numbers of the CO2 exchange, but as the water-side measurements are very limited and their representativity for the EC data highly questionable, this aspect of the paper does not bring much additional information compared to existing literature on air-sea CO2 exchange.*
We agree that the manuscript as it stands contains many components. Being the first paper on the Penlee Point Atmospheric Observatory, by necessity it needed to contain some detailed site descriptions and flux data validation. We tend to agree with Reviewer 2 that the paper is of appropriate length for ACP. However, for easier reading, we have restructured the paper slightly and moved the section on the theoretical flux footprint to an Appendix.

We are not in favor of removing the section on CO2 flux all together. Certainly more dissolved pCO2 measurements within the eddy covariance flux footprint would have been desirable. Without such measurements, we tried to verify our EC flux measurements with the best available information. Comparison between EC and predicted CO2 fluxes suggested that our measurements were within reason in terms of magnitude and direction. Additionally the EC measurements demonstrated high temporal variability in CO2 fluxes that are not captured by the weekly pCO2 measurements, which is of value. Finally, CO2 and CH4 were measured using the same instrument and the two gases were subject to the atmospheric turbulence. Thus the reasonable CO2 flux results gave us additional confidence in the CH4 flux measurements.

*The authors conclude that the site is suitable for long-term high temporal resolution measurements of air-sea exchange. For such a conclusion a much more thorough analysis is required. The site might be suitable for air-sea exchange representing coastal conditions, how representative the data are for undisturbed non-coastal air-sea exchange is not clear from the present analysis.*

We have made our statement more explicit in the revision and changed "air-sea exchange" to "air-sea exchange in shelf regions."

*Some more specific questions: Section 2.1: For EC measurements also the upwind topography is of importance, it is not clear from the text or figures how steep the topography is north of the site or how this might influence the measurements.*

Land northwest of the site slopes up at an angle of about 9°. In onshore airflow, the mean tilt angle is positive as air is forced upwards. The magnitude of this tilt for southwesterly wind, which blows perpendicularly across the Penlee headland and makes contact with water again to the northeast, is comparable to shipboard measurements.

We have zoomed out on the map in Fig.1 to show the topography more clearly:

[Figure]

Furthermore, a comparison of horizontal wind speed between Penlee and L4 when the wind is from the southwest does not show, within measurement uncertainties, a significant acceleration in the Penlee measurement (e.g. as might be expected when air is forced over a superstructure). So we don't believe the hill to the northwest of the site has a major influence on our measurements during southwesterly conditions.

*Section 2.2: The reason for the bias correction of the Windmaster Pro is very unclear and need to be explained.*
The manufacturer Gill describes this as a firmware 'bug.' We refer the reviewer to the now published technical note and will change the manuscript text accordingly: http://gillinstruments.com/data/manuals/KN1509_WindMaster_WBug_info.pdf

*Section 2.3: Is the signal dried in the Picarro (how does the low pressure of the Picarro exhaust give a dry signal)?*
Water vapor permeates through the membranes of the Nafion dryer, while $CO_2$ and $CH_4$ essentially do not. The sheath (i.e. outer) layer of the dryer is filled with the Picarro exhaust air, which is at a low pressure. Thus the pressure gradient between this outer layer and the inner tube (filled with the sample air at higher pressure) drives the water vapor out of the sample air.

*Page 11: How is the wave field influenced by the coast and how would this influence the drag coefficient?*
Thanks for the question. The bulk of the flux footprint at Penlee resides in waters ~20 m deep. Waves are considered to be in deep water if water depth is greater than half of the wavelength. They start to deviate significantly from deep-water behavior when the depth is less than about a quarter of the wavelength. At a wind speed of 10 m s$^{-1}$, fully developed wind waves have a wavelength of ~80 m. For wind speeds more than 10 m s$^{-1}$, wind waves near Penlee could be affected by depth, while swell (which tends to be longer) almost always would be. Thus PPAO should be considered a coastal, rather than a deepwater site. The exact effect of waves on the drag coefficient at Penlee is a topic of ongoing study and is beyond the scope of this paper.

*Page 13: SST measured very far away and there is no information at what depth the SST is measured. SST at this distance is probably not very representative for the flux footprint and this will most likely have a large impact on the bulk calculated sensible heat flux in Figure 5.*
SST was measured at the L4 station at approximately 1 m below the sea surface. It is true that the SST measurement at 6 km away might not always be representative of the Penlee flux footprint and this contributes to the apparent scatter in the EC vs. predicted heat flux comparison. We acknowledged this in the manuscript. Unfortunately there is no better alternative at this moment.

*Page 14: "different atmospheric dynamics" what does this refer to?*
Mainly the boundary layer dynamics. It is well known that the boundary layer height changes significantly over land, typically showing a maximum height during the day (due to solar heating of the surface) and a minimum height at night (due to longwave cooling). In comparison, the variability in boundary layer height is expected to be much less over the sea due to the thermal inertia of the ocean.

*Page 17: The 10 fold greater detection limit estimated by Peltola compared to the present study is explained by the higher variability over land than over sea. To me the variability over sea (in Figures 10 and 11) also seems relatively large.*

Thanks for the comment and sorry we didn't make this point clearer. Here we referred to periods of onshore wind only. As discussed in Section 4.2, ambient variability (1 standard deviation without the noise contribution, after detrending in 10-minute windows) in CH4 mixing ratio in marine air is as low as 0.2 ppb. In contrast, when the wind is from the north (i.e. from land), ambient variability in CH4 averages to ~3 ppb. In other words, CH4 mixing ratio is much more constant is marine air.

---

## Author Comment (AC2) · 18 Apr 2016

Author Comment with regard to:

**Air-Sea Fluxes of CO2 and CH4 from the Penlee Point Atmospheric Observatory on the South West Coast of the UK**

by M. Yang et al.

18 April, 2016

Many thanks for the thoughtful *comments and suggestions from the Anonymous Referee #2*. We are very glad to hear that the referee found our contribution valuable. Below we present each comment (in *italic*), followed by our reply. All of our replies are incorporated into the revised manuscript where appropriate, unless indicated otherwise.

**Summary and General Comments:** *Yang et al present the first eddy covariance measurements of air-sea exchange of methane, a significant advance in the field. The manuscript outlines the detection limit for the flux measurements using a Picarro cavity ringdown instrument. In addition, Yang et al present coincident CO2 flux measurements to make the argument that the Penlee Point Atmospheric Observatory (PPAO) is a uniquely situated for EC measurements of air-sea exchange in coastal regions.*
*In general, the paper is well written and is of appropriate length for ACP. The novel elements of this manuscript center around the CH4 EC flux assessment. I recommend the paper to be published after the authors attention to the following minor comments.*

**Specific Comments:**
*Page 1 Line 15: Perhaps define quantitatively "reasonable agreement" as this forms the foundation for the argument of PPAO as a site for future air-sea exchange measurements.*
Thanks for the suggestion. Judging mainly from the momentum transfer observations (and also the sensible heat flux comparison when the mast was fully raised), air-sea flux measurements at PPAO during southwesterly winds are in the mean within 20% of the open ocean air-sea transfer rates. This figure has been included in the revised manuscript.

*Page 1 Line 20: Why are the fluxes listed in order of 15, 27, 18 m.*
Chronologically the measurements were made at these mast heights. However, for easier reading we have listed the heights sequentially in the revision.

*Page 1 Line 23: I encourage the authors to remove the "~" for the detection limits. This should be a calculation where the detection threshold is defined for a specific averaging time, not an approximation. If there is unconstrained uncertainty, perhaps state an upper limit?*
Agreed. We have removed the '~' sign.

*Abstract: The abstract sells short the novel elements of the CH4 air-sea flux determinations. I encourage the authors to conclude the abstract with a more definitive statement that puts these new measurements in the content of what was known prior.*

Thanks for the suggestion. We have re-written the latter part of the abstract which now reads:

"We report, to the best of our knowledge, the first successful eddy covariance measurements of $CH_4$ emission from a marine environment. Higher sea-to-air $CH_4$ fluxes were observed during rising tides ($20\pm3$; $29\pm6$; $38\pm3$ µmole m$^{-2}$ d$^{-1}$ at 15, 27, 18 m AMSL) than during falling tides ($14\pm2$; $21\pm5$; $22\pm2$ µmole m$^{-2}$ d$^{-1}$, respectively), consistent with an elevated $CH_4$ source from an estuarine outflow driven by local tidal circulation. These fluxes are a few times higher than the predicted $CH_4$ emissions over the open ocean but are significantly lower than estimates from other aquatic $CH_4$ hot spots (e.g. polar regions, rivers and lakes). Finally, based on observations at PPAO, we found the detection limit of the eddy covariance $CH_4$ flux measurement to be 20 µmole m$^{-2}$ d$^{-1}$ over hourly timescales (4 µmole m$^{-2}$ d$^{-1}$ over 24 hours)."

*Page 3 Line 2: The abstract gives the impression that fluxes were determined at a range of altitudes at the same time, however it is clear here, that the range in altitudes also corresponds to a range in sampling periods. This should be noted in some fashion in the abstract?*

We have replaced the following sentence "Measurements from the southwest direction (background marine air) at three different sampling heights (approximately 15, 18, 27 m above mean sea level, AMSL) in three different periods during 2014–2015 are shown."

with..

"Measurements from the southwest direction (open water sector) were made at three different sampling heights (approximately 15, 18, 27 m above mean sea level, AMSL), each from a different period during 2014–2015."

*Page 4 Line 20: More detail on the bias correction should be included here. Is this a general result for all windmaster pro's? Is the value set by comparison to the G3? Or was this number from Gill?*

The values of this bias correction are recommended by Gill. The manufacturer describes this as a firmware 'bug.' We refer the reviewer to the now published technical note and have changed the manuscript text accordingly:
http://gillinstruments.com/data/manuals/KN1509_WindMaster_WBug_info.pdf

*Page 5 Line 1: Is the 5 slpm, 2m subsampling line still turbulent? If not, how important is this?*

Yes flow in the 1/4 inch outer diameter (1/8 inch inner diameter) subsampling line is turbulent. The Reynolds number inside is about 2400, which exceeds the threshold number of 2000 for turbulent flow.

*Page 5 Line 9: Why is "ambient mixing ratios" in quotations? Presumably this refers to ambient absolute humidity?*

The Picarro instrument reports volumetric mixing ratios of CO2 and CH4 with respect to 1) measured air and 2) dry air (i.e. measured air with the humidity numerically removed). We referred to the former (i.e. ambient mixing ratios) in quotations here because in our

setup the Picarro was sampling after a Nafion dryer, which removed 80-90% of the humidity already from the sample air stream.

*Page 12 Line 8: What is the future prospective for making continuous dissolved SW measurements of CH4 or other gases at this site? This seems critical for the future success of this site. Can this be done at L4?*
Thanks for the comment. The technology is still some ways off from continuous measurements of dissolved CH4 on the L4 buoy (however there is an in situ oxygen sensor on the buoy). It is fairly straight-forward to take discrete water samples from L4 and/or within the footprint of PPAO and analyze them later in the lab. In the future it might also be possible to install a semi-automated dissolved CH4 measurement system on Plymouth Marine Laboratory's research vessel *Quest*, which goes past PPAO and on to the L4 station weekly.

*Page 13 Line 9: Given that the primary focus of this paper is on air-sea exchange in the shelf region, I think it is most appropriate to state here that PPAO is " ... high temporal resolution measurements of air-sea exchange in shelf regions."*
Suggestion accepted. Thank you.